# Passive directional sub-ambient daytime radiative cooling

Bikram Bhatia [1], Arny Leroy[1], Yichen Shen[2], Lin Zhao[1], Melissa Gianello[1], Duanhui Li[3], Tian Gu[3], Juejun Hu[3], Marin Soljačić[2] & Evelyn N. Wang[1]

Demonstrations of passive daytime radiative cooling have primarily relied on complex and costly spectrally selective nanophotonic structures with high emissivity in the transparent atmospheric spectral window and high reflectivity in the solar spectrum. Here, we show a directional approach to passive radiative cooling that exploits the angular confinement of solar irradiation in the sky to achieve sub-ambient cooling during the day regardless of the emitter properties in the solar spectrum. We experimentally demonstrate this approach using a setup comprising a polished aluminum disk that reflects direct solar irradiation and a white infrared-transparent polyethylene convection cover that minimizes diffuse solar irradiation. Measurements performed around solar noon show a minimum temperature of 6 °C below ambient temperature and maximum cooling power of 45 W m$^{-2}$. Our passive cooling approach, realized using commonly available low-cost materials, could improve the performance of existing cooling systems and enable next-generation thermal management and refrigeration solutions.

[1] Department of Mechanical Engineering, Massachusetts Institute of Technology, Cambridge, MA 02139, USA. [2] Department of Physics, Massachusetts Institute of Technology, Cambridge, MA 02139, USA. [3] Department of Materials Science and Engineering, Massachusetts Institute of Technology, Cambridge, MA 02139, USA. Correspondence and requests for materials should be addressed to B.B. (email: bikram@mit.edu) or to A.L. (email: aleroy@mit.edu) or to Y.S (email: ycshen@mit.edu)

Cooling technologies are essential for refrigeration and thermal management applications. Existing cooling processes primarily rely on vapor compression and fluid-cooled systems despite their complexity and high cost. Passive cooling approaches such as atmospheric radiative cooling, relying on the high transparency of earth's atmosphere at mid-infrared (mid-IR) wavelengths, can lead to simple and low-cost refrigeration and cooling strategies that can augment existing thermal management solutions[1–4].

Passive atmospheric radiative cooling approaches take advantage of the spectral overlap of the radiative emission of terrestrial objects near ambient temperature and the transparent "atmospheric window" in the wavelength range from 8 to 13 μm[5–7]. This radiative access to the cold upper atmosphere through the atmospheric window has been exploited since ancient times to achieve cooling below ambient temperature during the night[8–13]. However, during the day, radiative cooling solutions have to mitigate solar irradiation (on the order of 1000 W m⁻²), which is an order of magnitude greater than the radiative cooling potential (on the order of 100 W m⁻²) and can impede any cooling. Several recent studies have investigated approaches that rely on spectrally selective surfaces that minimize absorption in the solar spectrum while maximizing emission in the mid-IR wavelengths[14–22]. However, this tightly constrained problem that requires negligible absorption in the solar spectrum and maximum emission in the mid-IR necessitates specialized photonic structures that are expensive and may not be easily accessible. Furthermore, previous work on passive atmospheric radiative cooling has primarily focused on spectral selectivity to enhance cooling performance without much regard to the possibility of angular radiative control. A few studies have investigated the advantages of directional control to radiative cooling[23–26] and have recently proposed angle-selective photonic structures[27,28] to enhance cooling performance. While the benefit of blocking direct-solar radiation was well understood and fixed solar reflectors were used in several daytime radiative cooling experiments[23,29], no previous demonstration has utilized the directional nature of incident solar irradiation to achieve sub-ambient radiative cooling.

This work describes a directional approach to achieve sub-ambient passive atmospheric cooling during the day. The method takes advantage of the angular confinement of the solar flux in the sky—completely blocking radiative exchange in the narrow direct-solar direction while allowing energy transfer in other directions. We demonstrate, theoretically and experimentally, that significant cooling below ambient temperatures is possible for emitters that are reflective (white) or absorptive (black) in the solar spectrum, despite the large incident solar flux. Energy balance modeling predicts that this approach has the potential to achieve temperatures as low as 20 °C below ambient and cooling powers as high as 89 W m⁻². Using a proof-of-concept setup, we measured temperatures as low as 6 °C below ambient and maximum cooling powers of 45 W m⁻² for a solar-white emitter and 30 W m⁻² for a solar-black emitter around solar noon. The experimental setup fabricated using low-cost readily available materials—polished aluminum, white polyethylene sheet, and commercially available paint—exhibits the simplicity and ease of implementation of the approach.

## Results

### Directional approach to daytime radiative cooling

Passive terrestrial daytime radiative cooling relies upon the spectral separation between the high atmospheric transmission at mid-IR wavelengths, coinciding with blackbody emission at ambient temperature, and solar irradiation. Figure 1a shows the incident solar spectrum and atmospheric transmission in the zenith direction as a function of wavelength[30]. Previous studies

primarily relied on spectrally engineered surfaces that maximize radiative emission in the atmospheric window, while reflecting the incident solar radiation[6,20,21,31–33]. However, achieving such spectral selectivity is challenging, particularly due to the large solar flux that needs to be rejected almost perfectly to prevent heating.

The angular confinement of solar irradiation in the sky enables a complementary approach to passive daytime radiative cooling. Figure 1b shows the normalized clear-sky short-wavelength radiance for a solar zenith angle of 40° that illustrates the solar irradiation contribution from different parts of the sky[34,35]. The plot also shows the angular atmospheric transmittance at a representative wavelength of 10.5 μm estimated using $\tau_{atm}(\lambda,\theta) = \tau_0(\lambda)^{1/\cos\theta}$ [10], where $\theta$ represents the zenith angle and $\tau_0(\lambda)$ represents the atmospheric transmittance in the zenith direction. In comparison with radiance due to the sun, which is concentrated around the solar disk, atmospheric transmittance is nearly constant across all angles other than near the horizon. This angular restriction of the solar irradiation in the sky relative to the broad angular range of high atmospheric transparency in the mid-IR provides an opportunity to selectively emit to the part of sky away from the sun and achieve passive cooling.

Figure 1c schematically shows a configuration that enables sub-ambient passive radiative cooling using a directional approach. The proposed concept comprises an emitter in thermal communication with the atmosphere and a reflector that blocks direct-solar radiation. The emitter is enclosed within a readily available cover that is partially transparent in the atmospheric window and partially reflective in the solar spectrum to minimize heat gain due to diffuse-solar radiation. The overall cooling power of the emitter (per area) $P_{cooling}$ at a temperature $T$ can be estimated by accounting for all contributions to the energy balance:

$$P_{cooling}(T) = P_{rad}(T) - P_{atm}(T_{amb}) - P_{solar\text{-}direct} - P_{solar\text{-}diffuse}$$
$$- P_{refl}(T_{refl}) - P_{cond\text{-}conv}(T, T_{amb})$$

$$(1)$$

The first term in Eq. 1, $P_{rad}$, represents the power radiated by the emitter. The second term, $P_{atm}$, represents the radiation emitted by the surrounding atmosphere at an ambient temperature $T_{amb}$ that is absorbed by the emitter. These contributions can be evaluated by integrating the spectral directional radiance leaving or absorbed by the emitter over all wavelengths and solid angles ($\Omega$) over the atmospheric hemisphere and excluding the solid angle subtended by the reflector ($\Omega_{refl}$) for $P_{atm}$, as shown in Eqs. 2 and 3.

$$P_{rad}(T) = \int_{\Omega} d\Omega \cos\theta \int_0^\infty d\lambda I_{BB}(T,\lambda)\tau_{cover}(\lambda,\theta)\varepsilon(\lambda,\theta) \quad (2)$$

$$P_{atm}(T_{amb}) = \int_{\Omega-\Omega_{refl}} d\Omega \cos\theta$$
$$\int_0^\infty d\lambda I_{BB}(T_{amb},\lambda)\varepsilon_{atm}(\lambda,\theta)\tau_{cover}(\lambda,\theta)\varepsilon(\lambda,\theta)$$

$$(3)$$

Here, $I_{BB}$ represents the spectral radiance of a blackbody, $\varepsilon(\lambda,\theta)$ represents the spectral directional emittance of the emitter, $\varepsilon_{atm}(\lambda,\theta) = 1 - \tau_{atm}(\lambda,\theta)$ represents the spectral directional emittance of the atmosphere, and $\tau_{cover}(\lambda,\theta)$ represents the spectral directional transmittance of the cover.

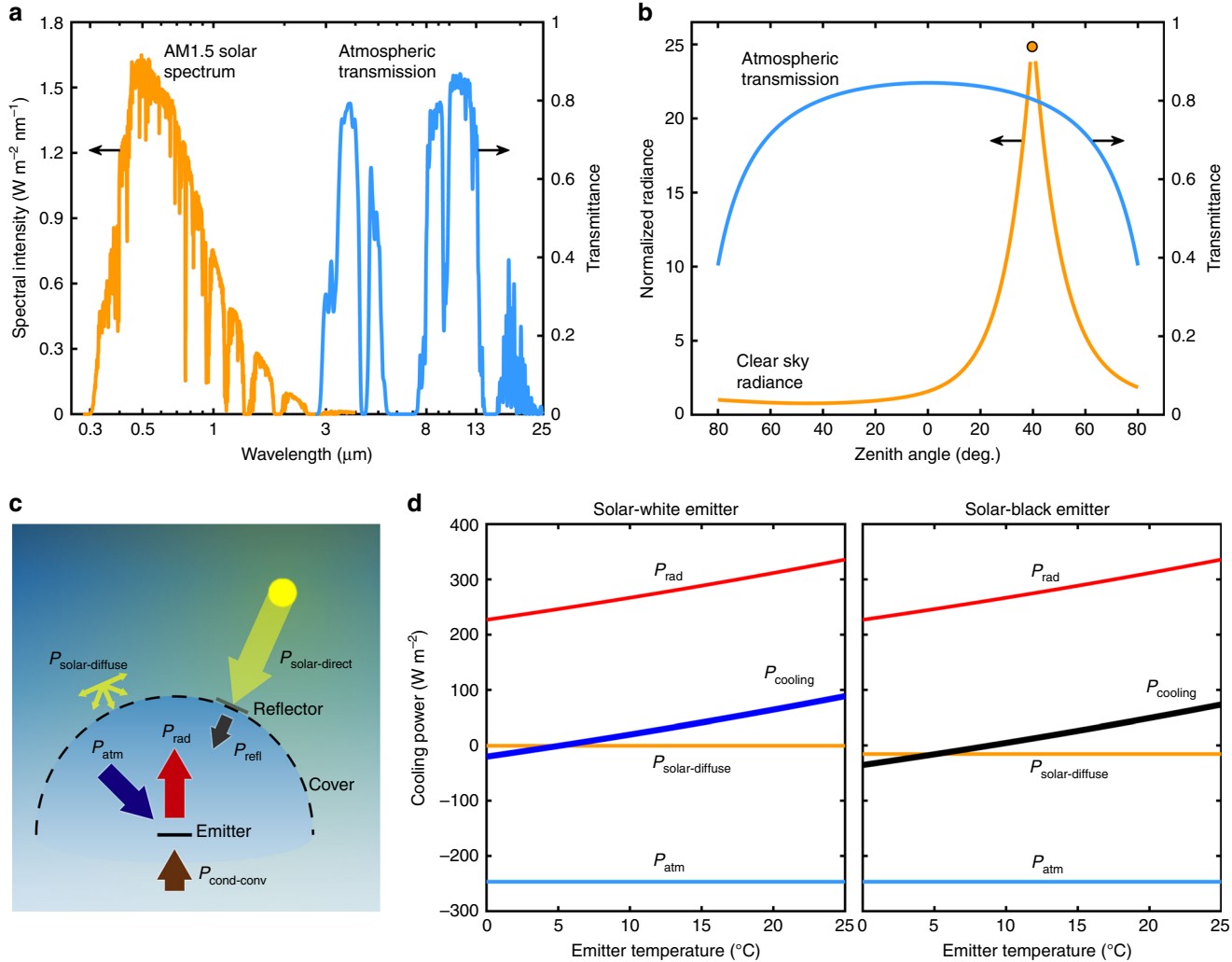

**Fig. 1** Passive directional daytime radiative cooling. **a** Spectral distribution of solar irradiation (AM1.5 G spectrum) and atmospheric transmittance (shown for wavelengths >2.7 μm, Cambridge, MA in October[30]). **b** Angular distribution of normalized clear sky radiance in a principal plane that includes the sun (denoted by the circle, shown for a solar zenith angle of 40°) and atmospheric transmittance (shown for 10.5 μm wavelength). **c** Energy flow diagram showing the possibility of achieving sub-ambient passive cooling during the day by emitting radiation in the mid-infrared wavelength range, while reflecting the angularly confined direct-solar radiation using a broadband reflector and an infrared-transparent cover that reflects diffuse-solar radiation. **d** Estimated net radiative cooling power $P_{cooling}$ as a function of emitter temperature (ambient temperature: 25 °C) and constituent contributions for an ideal solar-white emitter ($\lambda < 2.5$ μm: $\varepsilon = 0$, $\lambda \geq 2.5$ μm: $\varepsilon = 1$, $\forall \theta$) and ideal solar-black emitter ($\varepsilon = 1$, $\forall \lambda, \theta$) coupled with a perfect direct-solar reflector ($\rho_{refl} = 1$, $\forall \lambda, \theta$) and a representative diffuse-solar cover ($\lambda < 2.5$ μm: $\rho_{cover} = 0.8$, $\lambda \geq 2.5$ μm: $\tau_{cover} = 1 - \rho_{cover} = 0.9$, $\forall \theta$)

The incident solar irradiation comprises direct beam and circumsolar radiation emanating from the solar disk, equivalent to a solid angle of $6.87 \times 10^{-5}$ steradians (about 0.5° planar angle), and isotropic diffuse-solar radiation[36]. For the proposed configuration (Fig. 1c), the direct-solar irradiation, including the direct beam and circumsolar components, is rejected by the reflector and never reaches the emitter, that is $P_{solar-direct} = 0$. The contribution from the diffuse-solar radiation, $P_{solar-diffuse}$, transmitting through the cover and absorbed by the emitter is determined by estimating the isotropic diffuse-solar spectral radiance, $I_{solar-diffuse}(\lambda)$, as shown in Eq. 4. (Details of $I_{solar-diffuse}(\lambda)$ estimation are shown in Supplementary Note 1).

$$P_{solar-diffuse} = \int_{\Omega - \Omega_{refl}} d\Omega \cos\theta \int_0^\infty d\lambda I_{solar-diffuse}(\lambda)\tau_{cover}(\lambda, \theta)\varepsilon(\lambda, \theta)$$

(4)

The direct-solar reflector also emits radiation toward the emitter reducing its cooling power. The radiative contribution from the reflector toward the emitter cooling power $P_{refl}$, represented by Eq. 5, is dependent on the reflector emittance $\varepsilon_{refl}(\lambda, \theta)$ and temperature $T_{refl}$ (estimated using an energy balance on the reflector under direct-solar radiation). Thus the effect of the reflector can be minimal for a highly reflective surface or if the solid angle subtended by the reflector at the emitter is small.

$$P_{refl} = \int_{\Omega_{refl}} d\Omega \cos\theta \int_0^\infty d\lambda I_{BB}(T_{refl}, \lambda)\varepsilon_{refl}(\lambda, \theta)\tau_{cover}(\lambda, \theta)\varepsilon(\lambda, \theta)$$

(5)

In addition to the radiative contributions, conduction and convection from any support structure and surrounding air also

reduces emitter cooling. These non-radiative parasitic losses $P_{\text{cond-conv}}$ can be lumped together and quantified using an effective conductive-convective heat transfer coefficient $h_{\text{cond-conv}}$ as shown in Eq. 6.

$$P_{\text{cond-conv}} = h_{\text{cond-conv}}(T_{\text{amb}} - T) \qquad (6)$$

The potential cooling performance of the proposed approach is predicted using an idealized model based on the radiative contributions described above. Figure 1d shows the net cooling power and different radiative contributions for solar-white ($\lambda <$ 2.5 μm: $\varepsilon = 0$) and solar-black ($\lambda < 2.5$ μm: $\varepsilon = 1$) emitters with perfect emission in the infrared ($\lambda \geq 2.5$ μm: $\varepsilon = 1$) coupled with ideal direct-solar reflectors. The model assumes an easily available diffuse-solar cover with a typical solar reflectance of 0.8 and infrared transmittance of 0.9[37], and no parasitic heat gain (i.e., $h_{\text{cond-conv}} = 0$). At the 25 °C ambient temperature, $P_{\text{rad}} = 335.9$ W m$^{-2}$ and $P_{\text{atm}} = 246.7$ W m$^{-2}$ for both the solar-white and solar-black emitters, giving a total cooling potential of 89.2 W m$^{-2}$. The solar contribution depends on the magnitude of diffuse-solar radiation and emitter absorptance in the solar spectrum. Thus, for the presented case where the total $I_{\text{solar-diffuse}} = 76$ W m$^{-2}$, $P_{\text{solar-diffuse}} = 0.5$ W m$^{-2}$ for the solar-white emitter and $P_{\text{solar-diffuse}} = 15.6$ W m$^{-2}$ for the solar-black emitter. Overall, the model shows that a solar-white emitter can have a maximum cooling power of 88.7 W m$^{-2}$ and minimum temperature of 20 °C below ambient, while a solar-black emitter shows a maximum cooling power of 73.6 W m$^{-2}$ and minimum temperature of 16 °C below ambient. Even higher cooling powers and lower sub-ambient temperatures are possible using a diffuse-solar cover with

a higher solar reflectance and infrared transmittance. Thus we show that sub-ambient cooling is possible for a range of emitter properties using the proposed directional radiative cooling approach.

**Experimental design.** We designed a proof-of-concept demonstration that obstructed direct-solar irradiation, diminished diffuse-solar irradiation, maximized emission in the atmospheric window, reduced infrared absorption, and minimized heat gain due to conduction and convection. The device (Fig. 2) comprised of a thin, thermally conductive copper emitter (50 mm diameter) with its emitting surface coated using a commercially available white/black spray paint and back surface attached with a thermocouple. (Details of device design and fabrication are included in Supplementary Note 2). The emitter rested on thermal insulation (50 mm diameter) to minimize heat transfer due to conduction. Two layers of nanoporous polyethylene, separated by a 6.4 mm air gap, covered the emitter (while being physically separated) and minimized transmission of diffuse-solar radiation and served as a convection barrier. All lateral surfaces of the emitter-cover assembly were covered with aluminized Mylar and housed inside a polished aluminum cylinder and aperture (50 mm diameter) to minimize parasitic radiative heat transfer. A polished aluminum reflector (60 mm diameter), mounted on a custom-fabricated track, was suspended approximately 15 cm above the emitter plane to provide the emitter sufficient atmospheric access while keeping the device relatively compact. The path of the sun in the sky and its position at a given time determined the shape of the track and the reflector location relative to the emitter. The orientation of the device was

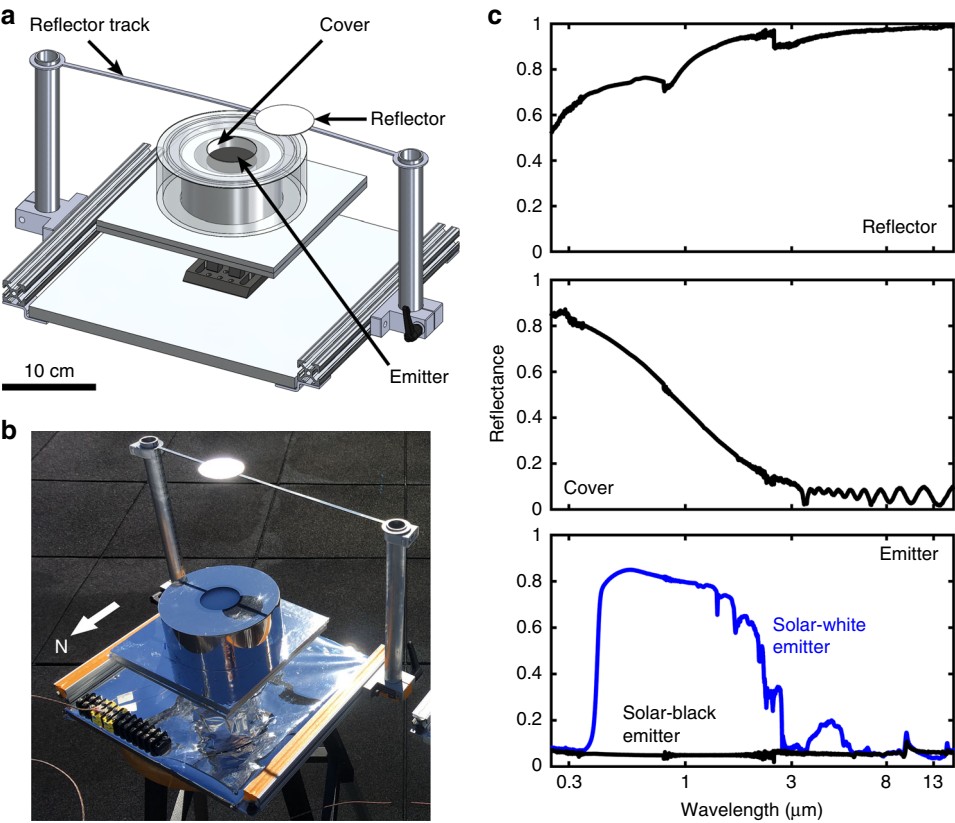

**Fig. 2** Proof-of-concept demonstration. CAD drawing (**a**) and photograph (**b**) of the fabricated device comprising of a white/black painted copper emitter that emits radiation in the mid-IR, a two-layer nanoporous polyethylene convection cover that partially reflects diffuse-solar irradiation, and a polished aluminum reflector capable of moving along a track that is adjusted based on the sun position and reflects direct-solar irradiation. **c** Spectral direct-hemispherical reflectance of the reflector (top), two-layer cover (middle), and white- and black-painted emitters (bottom)

determined based on the solar trajectory and the reflector was moved along the track manually during the course of the experiment.

The design of the experimental setup and spectral properties of the reflector and cover allowed decoupling the solar irradiation and mid-IR emission from the emitter, enabling passive daytime cooling. Figure 2c shows the spectral reflectance of the reflector, cover, and emitter(s) in the solar as well as the infrared wavelengths. (Additional emitter optical characterization results are included in Supplementary Figure 2). The polished aluminum reflector has broadband high reflectance and thus reflects most of the large direct-solar irradiation. While there is some absorption in the aluminum mirror due to its imperfect reflectance in the solar spectrum, cooling due to convection limits the temperature rise of the reflector. In addition, the small view factor between the reflector and emitter ensures minimal loss in emitter cooling power due to radiative transfer with the reflector. The double-layer nanoporous polyethylene convection cover, with a solar-weighted reflectance of 55% and an average transmittance of 92% in the atmospheric window, reflects a majority of the diffuse-solar irradiation while allowing transmission of almost all the radiation leaving the emitter. The paint-coated emitter has high emittance in mid-IR that maximized the emission in the atmospheric window. We chose two paints—one that was reflecting (white) and another that was absorbing (black) in the solar spectrum—to investigate the range of cooling performance as a function of emitter properties.

**Experimental results**. We performed outdoor measurements simultaneously on two devices placed next to each other, each comprising a polished aluminum direct-solar reflector, nanoporous polyethylene convection cover and painted copper emitter as described in the previous section. One device included an emitter coated with a solar-white paint while the emitter of the other device was coated with solar-black paint. (Details of the measurement setup are provided in Supplementary Note 4). To measure the lowest achievable temperature using our devices, we measured the stagnation temperature of the emitters on a clear day around solar noon (Fig. 3). (Refer Supplementary Figure 7 for the measured weather parameters for all experiments). Initially, the device apertures were covered to block atmospheric access as well as solar irradiation. Soon after the aperture covers were removed, the temperature of both the solar-white and solar-black devices dropped sharply and reached below the ambient temperature. At solar noon, the solar-white emitter reached a temperature of 6 °C below ambient and the solar-black emitter was 5.5 °C below ambient. While the solar-white emitter was always cooler than the solar-black, the difference in their temperatures was less than 1 °C, indicating that the contribution from solar absorption is small—likely from diffuse-solar irradiation. In addition, the emitter temperatures followed the ambient temperature trend closely and the temperature difference between the emitters and ambient increased after solar noon. These results can be attributed to parasitic heat gain due to conduction and convection, and solar absorption and heating of the exposed surfaces of the horizontally oriented device that decreased as the sun moved lower in the horizon beyond solar noon. Overall, the significant reduction of the device stagnation temperature, approximately 6 °C below the ambient temperature during the course of the measurement, demonstrates the possibility of achieving passive cooling using the demonstrated directional approach.

We also performed outdoor measurements to directly measure the cooling power as a function of emitter temperature. The cooling power measurement utilized an experimental setup and procedure similar to that for the stagnation temperature. Thin-film heaters were attached to the backside of both emitters, in addition to thermocouples, to quantify the cooling power at different emitter temperatures. We performed the measurement around solar noon on a mostly clear day (Fig. 4a). First, the emitters were allowed to passively cool below the ambient temperature as in the stagnation temperature measurement. Next, the PID-controlled heaters were turned on—the heater power was increased incrementally to raise the emitter temperature in approximately uniform steps until the emitter temperatures rose above the ambient temperature. Finally, we turned off the heaters and allowed the emitters to passively cool to their steady temperature below ambient. The input heater power, measured after the stabilization of emitter temperatures, for each step represents the passive cooling power of our system.

Figure 4b plots the time series data obtained (Fig. 4a) as cooling power as a function of emitter temperature for the solar-white and solar-black emitters. The maximum cooling power, corresponding to the measured power when the emitter and ambient temperatures are equal, was 45 W m$^{-2}$ for the solar-white emitter and 30 W m$^{-2}$ for the solar-black emitter. As expected, these values are lower than the cooling powers predicted by the idealized model shown in Fig. 1d, which assumed perfect emitter and reflector properties. The measured stagnation temperature, corresponding to zero cooling power, of the solar-white emitter was lower than the solar-black emitter by about 1 °C, as in the stagnation temperature measurement (Fig. 3). However, the maximum cooling below ambient temperature was lower than in Fig. 3, due to different atmospheric conditions and greater conductive thermal loss through the heater wires. Figure 4b also plots the corresponding modeled device cooling performance, which shows good agreement with experiment. The ideal model described earlier was modified to account for the measured spectral properties of the emitters, cover and reflector, device

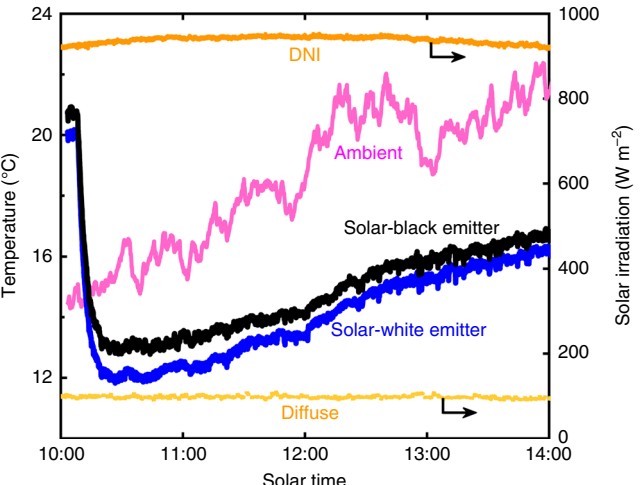

**Fig. 3** Stagnation temperature measurement around solar noon. Temperature of solar-white and solar-black emitters measured simultaneously 2 h before and 2 h after solar noon. Measured ambient temperature and direct normal irradiance (DNI) and diffuse-solar irradiation are also shown for reference. The nanoporous polyethylene cover shielded the emitters from diffuse-solar irradiation and the polished reflector was periodically moved along the track to prevent exposure from direct-solar irradiation. The devices were initially covered with aluminum covers, which were removed 5 min after starting data acquisition. Access to the atmosphere and reflection of solar irradiation caused the temperature of both devices to decrease drastically at first and then hold relatively steady approximately 5 °C below ambient temperature. The rooftop measurement was done on a clear day in Cambridge, MA (October 1, 2017)

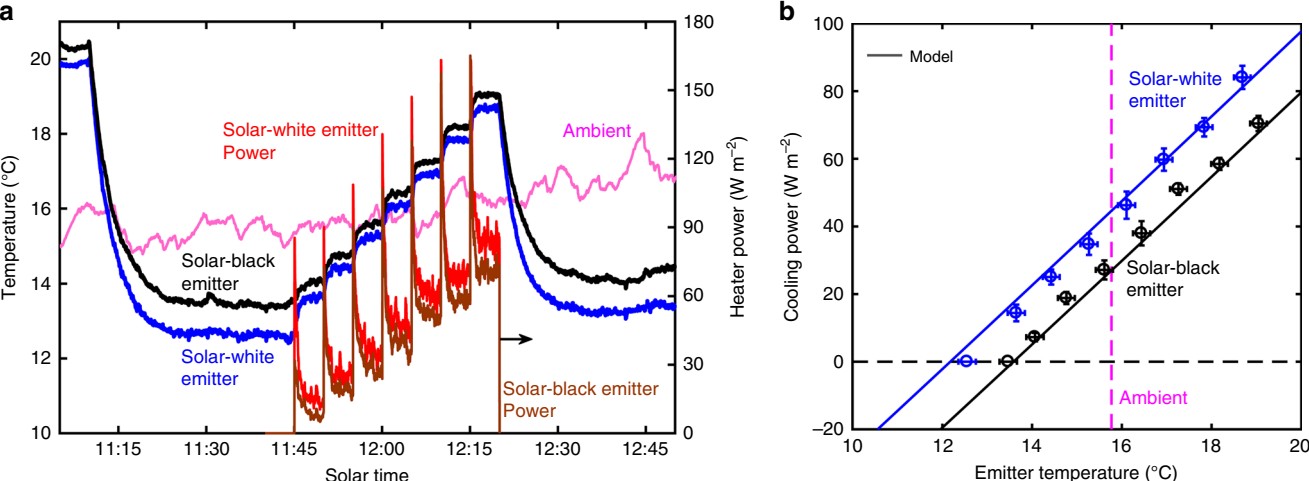

**Fig. 4** Cooling power measurement around solar noon. **a** Cooling power was measured using thin electrically insulating heaters attached to the back of the emitters. The heaters were initially off as the devices reached thermal equilibrium below ambient temperature, similar to the stagnation temperature measurement. Once the emitter temperature stabilized, the emitter temperature was raised above the ambient temperature in a step-wise manner by increasing the heater power (red and brown curves plotted on the right y axis, divided by the emitter area) regulated using PID control in 5 min increments. Finally, the heaters were turned off and the emitters allowed to reach stagnation temperature. **b** Cooling power measured for the solar-white and solar-black emitters as a function of emitter temperature. Each symbol corresponds to the heater power and emitter temperature at each step (shown in **a**), averaged over the last 2 min. Error bars represent the standard deviation of measured temperature and cooling power (details in Methods). Corresponding modeled performance calculated using measured properties and conditions is also shown. The constant ambient temperature value shown for reference represents the average ambient temperature measured during the power measurement. The measurement was done on a mostly clear day in Cambridge, MA (October 27, 2017)

geometry, and ambient temperature during the measurement, as well as the conductive–convective losses in the system. We quantified the conductive–convective loss using an effective heat transfer coefficient of $9.6\,\mathrm{W\,m^{-2}\,K^{-1}}$, estimated using a COMSOL model (Supplementary Note 5). The relatively high conductive–convective heat transfer coefficient indicates that better performance is possible—lower minimum temperatures and higher cooling powers at intermediate temperatures—through scale-up and improved thermal insulation. Maximum cooling power can also be increased by improving the radiative properties of the emitter, cover, and reflector and minimizing parasitic solar absorption by all surfaces.

## Discussion

This experimental demonstration of a novel directional approach to passive daytime radiative cooling provides a simple, low-cost method of achieving sub-ambient cooling. This approach takes advantage of the angularly confined nature of the dominant direct-solar irradiation to decouple it from the diffuse component, which is an order of magnitude lower in intensity. Unlike previous spectrally selective approaches that need to rely on near-perfect solar reflection to achieve sub-ambient cooling, our work demonstrates that it is possible to reach below ambient temperatures even with commonly available materials. In addition, by decoupling emission in the atmospheric window (by the emitter) and solar reflection (by the direct-solar reflector and diffuse-solar reflecting cover), we relax the optimization constraints that can lead to significantly improved cooling performance.

We believe this proof-of-concept demonstration is a significant first step that validates the concept of directional passive daytime radiative cooling and opens possibilities for improved device design and performance. One inherent constraint with the proposed directional approach is the need for sun position tracking. While the need for solar tracking prohibits infinite scaling of this concept, it is not necessarily limiting. Supplementary Note 6 shows an experimental measurement of stagnation temperature

using a band-type polished aluminum direct-solar reflector which ensured that the emitter was under shade and required no adjustment throughout the day. In addition, we used a white polyethylene cover made from a grocery bag, which had a solar-weighted reflectance of only 39% and transmittance of 67% in the atmospheric window. We measured a stagnation temperature of approximately 4 °C below ambient temperature—comparable to the performance reported in the Fig. 3 for a disk-type reflector, despite the larger solid-angle subtended by the band-reflector and sub-optimal radiative properties of the cover. Thus we envision a cooling device with an adjustable shadow ring-type direct-solar reflector, often used for diffuse sky radiation measurements[38,39], made using readily available low-cost materials.

This work could improve the performance of existing passive cooling solutions as well as lead to novel refrigeration and air-conditioning approaches. By eliminating the stringent require-ment to reflect direct-solar irradiation, even higher cooling power and lower temperatures can be achieved by combining the directional approach with existing spectrally selective surfaces for daytime radiative cooling—as demonstrated in past work[23,28]. In addition, this demonstration also proves the viability of future angular-selective photonic devices for passive daytime radiative cooling[27,28,40–42]. (The potential and possible applications of the directional daytime radiative cooling approach are further dis-cussed in Supplementary Notes 8 and 9). Furthermore, this directional radiative cooling can be readily implemented in thermal management solutions for concentrated photovoltaic systems, which already include solar-tracking systems[43,44]. Finally, a low-cost passive radiative cooler could enable safe storage of food and medicines in rural areas with limited access to electricity.

## Methods

**Temperature measurement**. Emitter temperatures were measured using K-type thermocouples (Omega 5TC-TT-K-36-36) attached on the back of the thin copper disk (near the center) using thermally conducting silver paste. All thermocouples were calibrated prior to application using a precise immersion-

style resistance temperature detector (RTD) sensor (Omega P-M-A-1/4-3-1/2-PS-12) and a chiller (Thermo Scientific A25). The RTD sensor and thermocouples were inserted into holes drilled in an isothermal copper block, which was immersed in the chiller water bath. The RTD temperature was read using a multimeter (Keithley 2000) and the thermocouples were read using a DAQ module (Measurement Computing USB-TC) with on-board cold junction compensation sensors enclosed in an aluminum box—similar to the configuration used for outdoor measurements. The calibration result for each thermocouple was used to correct the offset error and the slope error was propagated to calculate the measurement uncertainty (approximately ±0.2 °C), represented by the x-axis error bars shown in Fig. 4b.

**Cooling power measurement**. Cooling power was determined by measuring the electrical power input into Kapton® insulated flexible heaters (Omega KHR-2/2-P) attached to the back of the copper emitters. Each heater was connected to a sourcemeter (Keithley 2425) using a four-wire configuration and the input power was regulated by PID control implemented using LabVIEW. The sourcemeter accuracy and fluctuation in measured heater power (during the averaging period, after the initial sharp change in power) were used to calculate the cooling power uncertainty represented by the y-axis error bars shown in Fig. 4b. Previous studies have reported cooling power measured using PID control when the emitter temperature is equal to ambient temperature[21] or at different emitter temperatures by varying the fixed heater power and allowing the emitter temperature to respond based on thermal time constant of the device[18]. We measured the cooling power at different emitter temperatures using PID control, which allowed us to span the range of cooling powers at different operating conditions and perform measurements in a short time span (5 min per emitter temperature) when the weather conditions stayed relatively uniform.

**Solar-reflector tracking**. We computed the sun position (zenith and azimuth angle) relative to our experimental setup at the time and date of our experiment using an adapted version[45,46] of the solar position algorithm presented by Meeus[47]. The solar-reflector track path was then calculated from the computed sun position, for a fixed vertical distance from the emitter, so as to block the line of sight between the emitter and the sun during the time of the experiment. A reasonable vertical distance was chosen that would ensure a sufficiently small view factor between the emitter and the solar reflector (Supplementary Note 2). The solar-reflector track path was imported in a computer-aided design software to design the solar-reflector track. Finally, the track was cut from a 1.5 mm thick aluminum sheet by water jet.

**Optical property measurement**. We measured the direct-hemispherical reflectance of the reflector, polyethylene cover, and absorbers using a UV-Vis-NIR spectrophotometer (Cary 5000, Agilent) with an integrating sphere (Internal DRA-2500, Agilent) and an FTIR spectrometer (Nicolet 6700, Thermo Scientific) with an integrating sphere (Mid-IR IntegratIR™, Pike Technologies).

**Solar DNI and diffuse measurement**. The direct normal irradiance (DNI) and the global tilted irradiance (GTI) were measured by a pyrheliometer (EKO MS-56, ISO First Class) and a pyranometer (EKO MS-402, ISO First Class), respectively. Both sensors were mounted on a two-axis tracker (EKO STR-32G) and aligned to point to the sun during tracking. The pointing accuracy of the tracker was <0.01°. The diffuse-solar irradiance was calculated as the difference between GTI and DNI.

## Data availability
The data that support the findings of this study are available from the corresponding authors upon request.

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

## Acknowledgements

This work was supported by the Solid-State Solar Thermal Energy Conversion (S3TEC) Center, an Energy Frontier Research Center funded by the U.S. Department of Energy, Office of Science, Basic Energy Sciences under Award No. DE-SC0001299/DE-FG02-09ER46577. B. Bhatia acknowledges partial funding support from the Cooperative Agreement between the Masdar Institute of Science and Technology (Masdar Institute), Abu Dhabi, UAE and the Massachusetts Institute of Technology (MIT), Cambridge, MA, USA, Reference 02/MI/MIT/CP/11/07633/GEN/G/00. The authors thank George Ni and Jay Dhariwal for sharing weather monitoring equipment and data.

## Author contributions

B.B. and Y.S. conceptualized the project. M.G. and Y.S. contributed to the fabrication of early iterations of the experimental setup. B.B. and A.L. designed and conducted the experiments. B.B. and L.Z. performed material characterization. B.B. and L.Z. conducted theoretical modeling. D.L. and T.G. performed solar irradiation measurements. B.B. wrote the paper with input from all authors. J.H., M.S., and E.N.W. supervised and guided the project.

## Additional information

**Competing interests:** The authors declare no competing interests.

