## [Peer Review File · Nature Communications]

Reviewers' comments:

Reviewer #1 (Remarks to the Author):

This paper utilises angular control of incoming solar radiation to enhance the daytime cooling power available from a high emittance surface, whether black or white at solar wavelengths. Its scientific novelty is in its being one of very few contemporary experimental verifications that the combination of angular control of solar irradiance falling onto a surface while most emitted radiation still escapes can lead to a significant rise in net cooling rate and hence the extent to which surface temperatures can be reduced below ambient. The results are convincing. Their system is new approach but they are not the first, as stated in the paper, to experimentally demonstrate the value of a combined passive IR, solar angular selective approach. They are the first to do it via tracking. The other work was first done by Trombe in 1967 (who got further below ambient in a much hotter higher summer intensity incoming IR zone) and in papers and patents around that time. An experiment reported in an SPIE conference paper by Gentle et al in 2017 also had yet another novel approach to using the combination. Both studies must be acknowledged in this paper (refs below).

The tracking methodology utilised is new and does facilitate and clearly demonstrate extra net cooling. Its main drawbacks are partly explained and are two fold, the need for tracking the sun using a blocking disc, and to do that the need for ancillary complex systems. Are cooling gains enough to justify these? The science and approach is worth reporting, wide usefulness other than niche is doubtful but possible. Secondary issues concerned the reliance on beam solar radiation though some efforts to reduce diffuse solar heat gain are involved and help. A band variation involving limiting tracking over any one day is also worth consideration. However neither system is truly passive over longer periods than a day. Trombe and Gentle et al, and Smith (as cited) systems are in contrast fixed. It is an area of high current and recent interest so alternative methods like this are well worth reporting at this time and should have an impact.

The paper is well written, standard models are used, it is easy to read, and well presented. Reference list is fairly complete though those old and recent noted above should be added. Experimental details are spelled out in good detail for duplication, and are well supported by images. The modelling approach is acceptable for the task, though more on uncertainties and variances in the various parasitic, non-radiative heat flows involved would be of interest. What exactly was done to get errors and error bars in plots?

F. Trombe, Perspectives sur l'utilisation des rayonnements solaires et terrestres dans certaines régions du monde, Rev. Gén. Therm. 6 (1967) 1285-1314.

F. Trombe, US patent 3,310,102 (filed 1965) March 21, 1967 (includes several systems which part or fully block the sun plus enhanced convection suppression). Related experiments are in the paper that followed patent approval.

A. Gentle et al, Proc. SPIE 10369, Thermal Radiation Management for Energy Applications, 103690B (6 September 2017); doi:10.1117/12.2274568

Publication is recommended with mandatory additions to the reference list and optional extra commentary on the related approaches and data where radiation loss is largely maintained while solar beam radiation is blocked as mentioned above. I suggest in conclusion that they note that even higher cooling rates are feasible as shown in these other papers.

Reviewer #2 (Remarks to the Author):

Please see attached.

Reviewer #3 (Remarks to the Author):

The authors reported an experiment of achieving sub-ambient radiative cooling in the day using a directional approach. Different from most studies of achieving daytime radiative cooling using spectral selective emitters with near-complete reflection of sunlight and high emissivity in the atmosphere transparency window, the authors here used a small-area metal reflector to block the directional part of sunlight and used nanoporous polyethylene films to minimize absorption of diffuse sunlight. Such an approach minimized the requirement on the optical properties of the emitter over solar wavelength range, thus simplifying the optical design of the emitter. The demonstrated concept of directional radiative cooling is quite novel. The experimental results are backed up by modeling. The paper is generally well written.

However, a big concern is whether such an approach with the added complexity of requiring a reflector to block sunlight has advantage over radiative coolers based on spectral selectivity. I would suggest the following issues be addressed first before further consideration of this manuscript for publication in Nature Communications.

1. It is unclear whether this directional approach has advantage over cooling based on spectral selective emitters, in terms of cost and performance. For cooling based on spectral selective emitters, there have been quite simple demonstrated designs with good performance, such as Refs. 21 and 29. This study requires the use of a reflector to block the directional part of sunlight, which can inhibit scaling up and also adds complexity.
2. It would be useful for the authors to provide more details about the mentioned use of the directional radiative cooling in concentrated photovoltaic systems. Would the requirement of blocking the directional part of sunlight directly reduce the amount of sunlight reaching the photovoltaic systems? Also, would the diffuse cover used in this study cause a problem for concentrating sunlight?
3. The energy balance equation for the emitter in Eq. 1 misses a term of thermal radiation from the emitter to the reflector, which is the opposite process to Eq. 5.
4. Do the emissivities of the solar-white emitter and the solar-black emitter depend on angle? Does Eq. 3 require emitters having angle-independent emissivities? As the diffuse cover has diffuse transmission, Eq. 3 seems to only work when the emissivity is angle-independent.

Reviewer 1

REVIEWER

This paper utilises angular control of incoming solar radiation to enhance the daytime cooling power available from a high emittance surface, whether black or white at solar wavelengths. Its scientific novelty is in its being one of very few contemporary experimental verifications that the combination of angular control of solar irradiance falling onto a surface while most emitted radiation still escapes can lead to a significant rise in net cooling rate and hence the extent to which surface temperatures can be reduced below ambient. The results are convincing. Their system is new approach but they are not the first, as stated in the paper, to experimentally demonstrate the value of a combined passive IR, solar angular selective approach. They are the first to do it via tracking. The other work was first done by Trombe in 1967 (who got further below ambient in a much hotter higher summer intensity incoming IR zone) and in papers and patents around that time. An experiment reported in an SPIE conference paper by Gentle et al in 2017 also had yet another novel approach to using the combination. Both studies must be acknowledged in this paper (refs below).

The tracking methodology utilised is new and does facilitate and clearly demonstrate extra net cooling. Its main drawbacks are partly explained and are two fold, the need for tracking the sun using a blocking disc, and to do that the need for ancillary complex systems. Are cooling gains enough to justify these? The science and approach is worth reporting, wide usefulness other than niche is doubtful but possible. Secondary issues concerned the reliance on beam solar radiation though some efforts to reduce diffuse solar heat gain are involved and help. A band variation involving limiting tracking over any one day is also worth consideration. However neither system is truly passive over longer periods than a day. Trombe and Gentle et al , and Smith (as cited) systems are in contrast fixed. It is an area of high current and recent interest so alternative methods like this are well worth reporting at this time and should have an impact.

The paper is well written, standard models are used, it is easy to read, and well presented. Reference list is fairly complete though those old and recent noted above should be added. Experimental details are spelled out in good detail for duplication, and are well supported by images. The modelling approach is acceptable for the task, though more on uncertainties and variances in the various parasitic, non-radiative heat flows involved would be of interest. What exactly was done to get errors and error bars in plots?

F. Trombe, Perspectives sur l'utilisation des rayonnements solaires et terrestres dans certaines régions du monde, Rev. Gén. Therm. 6 (1967) 1285-1314.

F. Trombe, US patent 3,310,102 (filed 1965) March 21, 1967 (includes several systems which part or fully block the sun plus enhanced convection suppression). Related experiments are in the paper that followed patent approval.

A. Gentle et al, Proc. SPIE 10369, Thermal Radiation

Publication is recommended with mandatory additions to the reference list and optional extra commentary on the related approaches and data where radiation loss is largely maintained while solar beam radiation is blocked as mentioned above. I suggest in conclusion that they note that even higher cooling rates are feasible as shown in these other papers.

RESPONSE

We thank the reviewer for the positive assessment of our manuscript and providing useful suggestions to help improve the manuscript. Based on the reviewer's recommendations, we have made the following changes to the manuscript:

- The three references pointed out by the reviewer (F. Trombe, Rev. Gen. Therm 6, 1967; F. Trombe, US Patent 3310102, 1967; A. Gentle et al., Proc. SPIE 10369, 2017) are now included in the manuscript. These are certainly very relevant for the current manuscript. We thank the reviewer for informing us of these missing references. In addition, the literature review in the introduction has been revised to include more details about related atmospheric radiative cooling approaches that relied on blocking solar beam radiation (Revised manuscript, Page 3):

Furthermore, previous work on passive atmospheric radiative cooling has primarily focused on spectral selectivity to enhance cooling performance without considering the possibility of angular radiative control. A few studies have investigated the advantages of directional control to radiative cooling^{23–26} and have recently proposed novel angle-selective photonic structures^{27,28} to enhance cooling performance. While the benefit of blocking direct solar radiation was well-understood and fixed solar reflectors were used in several daytime radiative cooling experiments,^{23,29} no previous demonstration has utilized the directional nature of incident solar irradiation to achieve sub-ambient radiative cooling.

- We have also revised the conclusions to highlight that even higher cooling power and lower temperatures are possible using a directional approach to daytime radiative cooling, as demonstrated in past work. (Revised manuscript, Page 13):

By eliminating the stringent requirement to reflect direct solar irradiation, even higher cooling power and lower temperatures can be achieved by combining the directional approach with existing spectrally selective surfaces for daytime radiative cooling – as demonstrated in past work.^{23,28} In addition, this demonstration also proves the viability of future angular-selective photonic devices for passive daytime radiative cooling.^{27,28,40–42} (The potential and possible applications of the directional daytime radiative cooling approach are further discussed in Supplementary Sections 7 and 8).

We also added a detailed analysis of the potential of the directional radiative cooling approach and its comparison with existing spectrally selective methods in Supplementary Section 7.

- In order to address the concerns regarding the practical realization of this approach, we have added a section that provides examples of how the directional radiative cooling approach can

be implemented using a tracking configuration and “non-tracking” (or passive tracking) configuration. We also discuss the benefits and challenges of the directional radiative cooling approach (Supplementary Section 8).

- The COMSOL model was validated using additional experiments, modified to include the effect of wind velocity and various sources of parasitic, and non-radiative heat losses were analyzed (Supplementary Section 4).
- Detailed description of the methodology used to calculate the errors bars shown in Figure 4b was included in the methods sections:

(Revised manuscript, Methods, Temperature measurement, Page 14):

The calibration result for each thermocouple was used to correct the offset error and the slope error was propagated to calculate the measurement uncertainty ($\approx \pm 0.2$ °C), represented by the x-axis error bars shown in Figure 4b.

(Revised manuscript, Methods, Cooling power measurement, Page 15):

The sourcemeter accuracy and fluctuation in measured heater power (during the averaging period, after the initial sharp change in power) were used to calculate the cooling power uncertainty represented by the y-axis error bars shown in Figure 4b.

Reviewer 2

REVIEWER

Nature Communications Manuscript NC1714123, entitled “Passive directional sub-ambient daytime radiative cooling” by B. Bhatia et al.

In this manuscript, the authors explore the concept of angle-selective radiative cooling. Drawing inspiration from recent publications on the topic, the authors show that a polished aluminum disk can be used to block solar illumination, given proper solar tracking. Under ambient conditions outdoors, solar radiation is blocked enough to create a small temperature difference between black and white emitting film-based radiative coolers. In both cases, temperatures are about 5.5-6 K below ambient temperatures. The authors also apply heating to estimate that the cooling power of black and white radiative coolers operating at ambient temperatures are 30 and 47 W/m², respectively.

Overall, this manuscript is a potentially significant realization of a recently-developed theoretical concept. Nonetheless, several issues arose in reviewing this manuscript that should be addressed before potential publication:

RESPONSE

We thank the reviewer for evaluating our manuscript and providing useful comments. We address each issue raised by the reviewer in our responses below. Based on the reviewers’ suggestions, we performed additional measurements, revised modeling and included additional information to

improve the manuscript. We hope these changes satisfactorily address the concerns of the reviewer.

REVIEWER

1. Given that the emission spectra of the black and white emitting films (Fig. 1), and their observed relatively close temperatures under outdoor ambient conditions in Cambridge, MA (Fig. 3), it is somewhat unclear exactly why their cooling power would differ by more than 50% (Fig. 4). Can any additional detail / discussion on this point be provided?.

RESPONSE

We appreciate the excellent question. To determine the reason, we compared different cooling power contributions for the solar-white and solar-black emitters using the model (which matched the experimental results, as shown in Figure 4). The reason for the relatively small (~ 1 °C) difference in stagnation temperatures of the solar-white and solar-black emitters is the large conductive-convective heat transfer coefficient ($9.6 \text{ W/m}^2\text{K}$) in our experiment.

Figure R1 below compares the breakdown of different contributions for two different heat transfer coefficients: $h = 9.6 \text{ W/m}^2\text{K}$ (used for the experiment) and $h = 1 \text{ W/m}^2\text{K}$ (hypothetical lower value). All the radiative contributions to the net cooling power, P_{cooling} : P_{rad} , P_{atm} and $P_{\text{solar-diffuse}}$, are a function of the emission spectra shown in Figure 2c. On the other hand, the conductive-convective loss contribution, $P_{\text{cond-conv}}$, is independent of the spectra. When the emitter temperature is equal to the ambient temperature (represented by the magenta vertical dashed lines), the conductive-convective contribution is zero and the maximum cooling power is only a function of the radiative contributions. When h is small (Figures R1a and b), the relative contribution of conductive-convective heat transfer at the stagnation temperature (represented by the black vertical dashed line) is small in comparison to other radiative contributions and a large difference in the stagnation temperature is observed between the solar-white (Figure R1a, stagnation temperature: 3.7 °C) and solar-black emitter (Figure R1b, stagnation temperature: 8.2 °C). However, when h is large – as is the case in our experiment (Figures R1c and d) – the relative contribution of the conductive-convective heat transfer becomes significant and is greater for the solar-white emitter (which has larger $T_{\text{amb}} - T_s$) than the solar-black emitter at their respective stagnation temperatures (Figure R1c, solar-white emitter, stagnation temperature: 12.2 °C; Figure R1d, solar-black emitter, stagnation temperature: 13.5 °C). Thus, we observed relatively close stagnation temperatures for the solar-white and solar-black emitters in our experiments.

Figure R1. Breakdown of different contributions to the net cooling power of solar-white and solar-black emitters when the heat transfer coefficient is $1 \text{ W/m}^2\text{K}$ (**a** & **b**) and $9.6 \text{ W/m}^2\text{K}$ (**c** & **d**). The magenta vertical dashed line represents the ambient temperature and black vertical dashed line represents the stagnation temperature at which net cooling power, $P_{\text{cooling}} = 0$. The small cooling power contribution from the reflector ($<2 \text{ W/m}^2$) is not shown in the plots for clarity.

REVIEWER

2. The authors estimate a single convective coefficient of $9.6 \text{ W/m}^2\cdot\text{K}$. However, convection generally depends on wind speed, and thus may vary over the course of the four hours over which data is taken (Fig. 3). It would be helpful to re-calculate the convective coefficient as a function of time and to ideally plot it, or at least to present the range and variance of this value, and calibrate

it against the corresponding variation in wind speeds during that time, to help with interpretation of the results, as well as to aid future experimentalists in designing their setup.

RESPONSE

We thank the reviewer for this question. We did originally neglect the effect of variation of wind speed in our conduction-convection heat transfer coefficient estimation. For the COMSOL model, used to evaluate the heat transfer coefficient, we chose a wind speed of 1 m/s. But, we have revised the modeling as per the reviewers' suggestion. Figure R2a below shows the effect of evaluated heat transfer coefficient as a function of wind velocity. As shown, the conductive-convective heat transfer coefficient for our device is relatively insensitive to the wind speed beyond the initial increase from 0 to 1 m/s. We believe this relative independence of the heat transfer coefficient from the wind speed is due to the closed structure of our device – it is surrounded by a rigid aluminum radiation shield on all sides except the emitter aperture, where it is covered by two-layers of polyethylene sheets (with an air gap in between).

Based on the reviewers' recommendation, we have included the effect of wind speed in the heat transfer coefficient evaluation which is used as an input parameter in the model used to estimate the cooling power (shown in Figure 4). The cooling power measurement was done over a half-hour period around solar noon. While we did not have reliable wind speed data from the weather station next to the experimental setup, we were able to find the wind speed for the time of our measurement from the Wunderground weather history at a location close to the site of the experiment (0.2 miles away). Figure R2b shows this weather history annotated with the time of the experiment. The weather data shows that the wind speed is about 17 mph (7.6 m/s) and does not change significantly during the course of the experiment. In the revised Supplementary Section 4, we now include the wind speed information used for modeling:

Other external boundary conditions were defined using convection correlations with respect to the ambient temperature – we used a constant wind speed around 7.6 m/s based on the conditions during the experiment.

Figure R2. a, Weather data, annotated with the duration of the cooling power measurement (shown in Figure 4). **b**, Modeled conductive-convective heat transfer coefficient as a function of wind speed.

REVIEWER

3. In Supplemental Material Section 4, the authors present a model of the device temperature distribution. Was any IR image taken to check the baseline accuracy of this model? If not, was there any experiment to validate the overall temperature model of the setup?

RESPONSE

We did not have direct validation of the temperature distribution between the model and experiment earlier. However, based on the reviewer's suggestion, we performed additional experiments and validated the COMSOL model presented in Section 4 of the Supplementary Information. It was not possible to take an IR image of our setup because of its closed structure and reflecting outside surfaces, but we installed six additional thermocouples at different locations within the device to capture the temperature distribution. We performed rooftop measurements under three different conditions and compared the experimental and modeled temperatures. The results show good agreement between the experiments and model predictions.

Details of the model validation are now added to Section 4 of the Supplementary Information, as show below:

In order to experimentally validate the temperature distribution predicted using the COMSOL model, we installed six additional thermocouples at different locations within the device with the solar-white emitter. We then performed three outdoor measurements (each 1 hour long), following a procedure similar to that used to obtain results shown in Figure 3, under different solar irradiation and weather conditions. Figure S5 shows the experimentally measured average temperatures for each of the three experiments (averaged after reaching thermal equilibrium). Figure S5 also shows the temperature distribution and temperatures evaluated at the thermocouple locations using the COMSOL model with the corresponding experimental conditions as the input. The modeled temperature distribution within the device shows good agreement with experimental results under different conditions. Minor deviations between experiments and model results can be attributed to thermocouple accuracy, variability in positioning the thermocouples and optical properties of different surfaces. Overall, the agreement between experimental and modeling results validates the COMSOL model used to estimate the conductive-convective heat transfer coefficient.

Figure S5. Model validation. Comparison of the experimentally measured temperatures at different locations of the device (left) with the temperature distribution predicted using the COMSOL model (right). Results are shown for three different conditions: (a) 1.5 hours after solar noon on a mostly clear day with scattered clouds (average ambient temperature: 26.8 °C), (b) clear night (average ambient temperature: 20.1 °C), and (c) around solar noon on a hazy day with some clouds (average ambient temperature: 29.0 °C). Measurements were done on the rooftop in Cambridge, MA between 2 pm August 23, 2018 and 2 pm August 24, 2018.

REVIEWER

4. In the discussion of future work, the authors mention that solar trackers are already present in concentrating photovoltaic (CPV) systems. However, usually they're designed to collect as much light as possible – the opposite of what's needed here. A brief comment on how these could be combined should help illuminate the authors' concept more clearly.

RESPONSE

We thank the reviewer for this helpful suggestion. Based on the reviewers' advice, we have added a supplementary section that outlines our vision of how directional radiative cooling might be

implemented in representative concentrated photovoltaic (CPV) systems and portable cooling applications. Specifically, we highlight that it should be possible to combine directional radiative cooling with the CPV systems without affecting light collection. Figure S8b (shown below) presents a concept drawing of a CPV system with a physically-separated reflector behind the PV cell. Since the CPV system always tracks the sun-position, the backside of the PV cell (which does not contribute to electricity generation) is always shaded from direct solar radiation but allows atmospheric access which enables radiative cooling.

The new section that provides more details about possible applications of directional radiative cooling included in the revised supplementary information is as follows:

Supplementary Section 8: Directional daytime radiative cooling – Applications

The directional approach to passive daytime radiative cooling presented in this work could be useful for applications such as portable refrigeration and concentrated photovoltaic (CPV) cooling. Figure S9 shows conceptual drawings of how the directional approach might be implemented to achieve passive cooling in a portable cooler and CPV system in a simple and cost-effective manner. Figure S9a presents one possible embodiment of a portable cooling device that comprises of a band-type reflector that can reflect direct solar radiation. The angular position of the band reflector will depend on the position of the sun in the sky and can be adjusted for a given day. The top surface of the insulated cooler comprises of the mid-IR emitter which has access to the atmosphere and could be covered by a solar-reflecting convection cover which could enable sub-ambient cooling.

Figure S9b demonstrates how directional radiative cooling might be implemented in a CPV system. A CPV system typically comprises of dual-axis solar-tracking concentrating optics with the PV cell at its focal point.^{5,6} Since the focal axis of the CPV always points towards the sun, direct solar radiation impinging on the back of the PV cell (which does not contribute to electricity generation) can be shaded by a fixed, physically-separated reflecting surface as demonstrated in the present study. The backside of the PV cell should be emitting in mid-IR wavelengths to allow atmospheric radiative cooling and can be solar-reflecting to minimize diffuse-solar absorption to further improve cooling performance. A separate convection cover is not necessary since CPV cells do not require sub-ambient cooling – in fact, convection due to wind should supplement cooling by radiation. Thus it should be possible to achieve radiative cooling in CPV systems using a directional approach by utilizing the existing solar-tracking configuration and without affecting electricity generation.

Figure S9. Possible applications. Concept drawings showing possible approaches for practical implementation of directional daytime radiative cooling in portable refrigeration (a) and concentrated photovoltaic systems (b).

REVIEWER

In conclusion, the authors have studied the effects of blocking the solar disc on radiative cooling, and shown some promising results for the resulting cooling power. If they can successfully address all the points listed above, then it may very well be appropriate to publish this manuscript in Nature Communications.

RESPONSE

We hope the reviewer will find the above responses and changes in the revised manuscript satisfactory. We thank the reviewer again for the constructive feedback which helped us significantly improve the manuscript.

Reviewer 3

REVIEWER

The authors reported an experiment of achieving sub-ambient radiative cooling in the day using a directional approach. Different from most studies of achieving daytime radiative cooling using spectral selective emitters with near-complete reflection of sunlight and high emissivity in the atmosphere transparency window, the authors here used a small-area metal reflector to block the directional part of sunlight and used nanoporous polyethylene films to minimize absorption of diffuse sunlight. Such an approach minimized the requirement on the optical properties of the emitter over solar wavelength range, thus simplifying the optical design of the emitter. The demonstrated concept of directional radiative cooling is quite novel. The experimental results are backed up by modeling. The paper is generally well written.

However, a big concern is whether such an approach with the added complexity of requiring a reflector to block sunlight has advantage over radiative coolers based on spectral selectivity. I would suggest the following issues be addressed first before further consideration of this manuscript for publication in Nature Communications.

RESPONSE

We thank the referee for reviewing our work and providing helpful comments. Based on the feedback from the reviewer, we have included in the revised manuscript quantitative comparison of the cooling performance of the directional and spectral approaches to daytime radiative cooling. In addition, we have added a section in the manuscript that discusses the practical implementation in real systems, including in CPV and portable cooling devices. The revised manuscript also validates the model and corrects the error pointed out by the reviewer. Please see below for a detailed discussion about these changes in the revised manuscript and point-by-point responses.

REVIEWER

1. It is unclear whether this directional approach has advantage over cooling based on spectral selective emitters, in terms of cost and performance. For cooling based on spectral selective emitters, there have been quite simple demonstrated designs with good performance, such as Refs. 21 and 29. This study requires the use of a reflector to block the directional part of sunlight, which can inhibit scaling up and also adds complexity.

RESPONSE

We thank the reviewer for highlighting this important aspect that was not addressed in the original manuscript. The revised manuscript includes a new section that analyzes the potential of the passive directional daytime radiative cooling and compares its performance with the existing method using spectrally selective emitters highlighted by the reviewer. Using the modeling framework presented in the paper and representative emitter properties, we show that while the spectrally selective emitters can reach a lower minimum temperature, the directional approach results in a higher cooling power during the day. This is primarily a result of how direct solar radiation is rejected in the two cases. In the spectrally selective case, even a relatively high solar reflectance of 0.96 can result in a ~ 40 W/m² reduction in maximum cooling power. On the other hand, for the directional approach, the negative contribution of the direct solar radiation is negligible. Additionally, because the directional approach blocks a narrow angular window (because of the angular confinement of the solar disk in the sky) in comparison with the spectral window blocked by typical selective emitters used for radiative cooling (Figure S8 below), the net outgoing cooling power is higher for the directional approach. But perhaps the greatest potential of the current work lies in the combination of selective emitters with the directional approach to achieve higher cooling power and lower stagnation temperatures during the day.

Challenges related to the scalability of the directional method certainly exist for large-scale installations, however the promise of better cooling performance could make it an attractive approach for space-limited applications such as CPV cooling and portable refrigeration.

The new supplementary section highlighting the potential performance benefits of directional daytime radiative cooling is shown below:

Supplementary Section 7: Directional daytime radiative cooling – Potential

This section explores the potential benefits of the directional approach to passive daytime radiative cooling, compares its performance with existing method using spectrally selective surfaces, and the possibility of combining the two to achieve significantly improved performance. To do so, we evaluated the cooling performance (using the modeling framework shown in the main text) for representative spectrally selective and directional configurations. Figure S8a shows the spectral emissivity of the emitters chosen for this analysis. For the “Spectral” approach, we used a square-wave-type spectrally selective emitter with an emissivity of 0.90 in the transparent atmospheric spectral window and a reflectivity of 0.96 (emissivity of 0.04) at all other wavelengths, including in the solar spectrum – representative of existing literature on spectrally selective surfaces for daytime radiative cooling.^{2–4} For the “Directional” approach, we used a step-function-type partly-solar-reflecting emitter with a reflectivity of 0.80 (emissivity of 0.20) in the solar spectrum (<2.5 μm) and an emissivity of 0.90 at longer wavelengths, including in the transparent atmospheric window – similar to the solar-white emitter used in the current study. In addition, we assumed a direct-solar reflector with a reflectivity of 0.96 at all wavelengths. No diffuse-solar-reflecting cover was considered for the ease of comparison.

Figure S8b compares the cooling performance of the different cases, assuming only radiative contributions (i.e., $h_{\text{cond-conv}} = 0$). We observe that while the Selective approach results in a lower emitter temperature, the Directional approach can achieve a significantly higher maximum cooling power than the Selective approach. The higher cooling power is primarily a result of how the direct solar radiation is rejected in the two cases. In the Selective case, the 0.96 solar reflectance translates to a direct reduction of cooling power during the day by $\sim 40 \text{ W/m}^2$. On the other hand, there is practically no cooling power penalty from the direct solar radiation in the Directional case. In addition, because of the confinement of the solar disk in the sky, the angular window blocked in the Directional approach to reject direct sunlight is significantly smaller than the spectral window blocked by square-wave-type emitters in the Selective approach resulting in a higher net outgoing cooling power during the day (and night). We also present a “Selective+Directional” case where the Selective emitter is combined with the Directional configuration with a broadband reflector to reject the direct-solar radiation. This combined configuration provides the best performance – high maximum cooling power and significantly lower minimum achievable temperature. These results indicate that the directional approach to daytime radiative cooling and its possible combination with current approaches using spectrally selective surfaces could improve passive daytime radiative cooling performance.

Figure S8. Cooling performance comparison. **a**, Emitter spectral properties used to compare the daytime radiative cooling using selective and directional approaches. **b**, Modeled cooling power as a function of emitter temperature (ambient temperature: 25 $^{\circ}\text{C}$) during the day for a solar-reflecting square-wave-type Selective emitter, partly solar-reflecting step-function-type emitter for the Directional approach, and the square-wave-type selective emitter coupled with the directional approach to reject direct sunlight for the Selective+Directional case.

REVIEWER

2. It would be useful for the authors to provide more details about the mentioned use of the directional radiative cooling in concentrated photovoltaic systems. Would the requirement of blocking the directional part of sunlight directly reduce the amount of sunlight reaching the photovoltaic systems? Also, would the diffuse cover used in this study cause a problem for concentrating sunlight?

RESPONSE

We thank the reviewer for this useful suggestion. Based on the reviewers' recommendation, we have added a supplementary section that provides more details (including concept drawings) about the implementation of directional radiative cooling in portable cooling and concentrated photovoltaic (CPV) systems. Specifically, we highlight that directional radiative cooling can be implemented in CPV systems without affecting the amount of sunlight reaching the PV cell. Since the focal-axis of a tracking CPV system is always collinear with direct-solar radiation and the PV

cell is placed at the focal point of the concentrating optics (Figure S9b, shown below), a fixed physically-separated reflector can be located at the backside of the PV cell. The backside of the PV cell is thus shaded from direct solar radiation and can emit in mid-IR wavelengths. The diffuse solar radiation can be reduced by ensuring that the backside of the PV cell is solar-reflecting (similar to the “solar-white” emitter used in this work). A separate convection cover is not necessary since CPV cells typically do not require sub-ambient cooling. In fact, convection due to wind should supplement radiative cooling which could help achieve a lower PV cell temperature.

More details about possible applications of directional radiative cooling are now included in a new section in the supplementary information as shown below:

Supplementary Section 8: Directional daytime radiative cooling – Applications

The directional approach to passive daytime radiative cooling presented in this work could be useful for applications such as portable refrigeration and concentrated photovoltaic (CPV) cooling. Figure S9 shows conceptual drawings of how the directional approach might be implemented to achieve passive cooling in a portable cooler and CPV system in a simple and cost-effective manner. Figure S9a presents one possible embodiment of a portable cooling device that comprises of a band-type reflector that can reflect direct solar radiation. The angular position of the band reflector will depend on the position of the sun in the sky and can be adjusted for a given day. The top surface of the insulated cooler comprises of the mid-IR emitter which has access to the atmosphere and could be covered by a solar-reflecting convection cover which could enable sub-ambient cooling.

Figure S9b demonstrates how directional radiative cooling might be implemented in a CPV system. A CPV system typically comprises of dual-axis solar-tracking concentrating optics with the PV cell at its focal point.^{5,6} Since the focal axis of the CPV always points towards the sun, direct solar radiation impinging on the back of the PV cell (which does not contribute to electricity generation) can be shaded by a fixed, physically-separated reflecting surface as demonstrated in the present study. The backside of the PV cell should be emitting in mid-IR wavelengths to allow atmospheric radiative cooling and can be solar-reflecting to minimize diffuse-solar absorption to further improve cooling performance. A separate convection cover is not necessary since CPV cells do not require sub-ambient cooling – in fact, convection due to wind should supplement cooling by radiation. Thus it should be possible to achieve radiative cooling in CPV systems using a directional approach by utilizing the existing solar-tracking configuration and without affecting electricity generation.

Figure S9. Possible applications. Concept drawings showing possible approaches for practical implementation of directional daytime radiative cooling in portable refrigeration (a) and concentrated photovoltaic systems (b).

REVIEWER

3. The energy balance equation for the emitter in Eq. 1 misses a term of thermal radiation from the emitter to the reflector, which is the opposite process to Eq. 5.

RESPONSE

We thank the reviewer for pointing out this error. Indeed, our model did not include the outgoing radiation contribution from the emitter in the direction of the reflector (contribution opposite of Equation 5, as the reviewer correctly pointed out). This contribution can be accounted for in the term P_{rad} , which now represents all power radiated by the emitter, by integrating the spectral directional radiance over the entire the atmospheric hemispherical solid-angle (including the reflector solid angle). The revised manuscript rectifies the error by correcting Equation 2 and corresponding description of the modeling framework:

The first term in Equation 1, P_{rad} , represents the power radiated by the emitter. The second term, P_{atm} , represents the radiation emitted by the surrounding atmosphere, at an ambient temperature T_{amb} , that is absorbed by the emitter. These contributions can be evaluated by integrating the spectral directional radiance leaving or absorbed by the emitter over all wavelengths and solid angles (Ω) over the atmospheric hemisphere, and excluding the solid angle subtended by the reflector (Ω_{refl}) for P_{atm} , as shown in Equations 2 and 3.

$$P_{\text{rad}}(T) = \int_{\Omega} d\Omega \cos \theta \int_0^{\infty} d\lambda I_{\text{BB}}(T, \lambda) \tau_{\text{cover}}(\lambda, \theta) \varepsilon(\lambda, \theta) \quad (2)$$

While rectifying this error was important to correctly capture the energy flow in our proposed configuration, the corresponding change in the model did not affect the overall results significantly. This is primarily due to the small view factor between the reflector and emitter (~ 0 for the idealized model shown in Figure 1 and 0.017 for the experimental configuration shown in Figure 2).

We have revised the model and updated the results in Figures 1d and 4b, and corresponding values in the manuscript.

We are really grateful that the reviewer caught this error and that we had an opportunity to correct it here.

REVIEWER

4. Do the emissivities of the solar-white emitter and the solar-black emitter depend on angle? Does Eq. 3 require emitters having angle-independent emissivities? As the diffuse cover have diffuse transmission, Eq. 3 seems to only work when the emissivity is angle-independent.

RESPONSE

The reviewer is correct in pointing out that Equation 3 assumes angle-independent emissivity for modeling the solar-white and solar-black emitters used in the experiments. In order to verify this assumption, we measured the angle-dependent emissivity of the emitters. The results indicate that the emissivity of both the solar-white and solar-black emitter is angle-independent for all angles relevant for radiative transfer within the experimental setup. Furthermore, the measured angle-dependent emissivity values are consistent with the direct-hemispherical measurement results shown in Figure 2c. Thus the angle-independent emissivity assumption should be valid in our case. The spectral-angular emissivity measurement results are now included in a new section in the supplementary information and shown below:

Supplementary Section 3: Spectral-Angular emissivity of the emitters

Figure S2 shows the measured spectral and angular emissivity of the solar-white and solar-black emitters used for the experiments. The angle-dependent emissivity was evaluated from reflectance measurements done using an FTIR spectrometer (Nicolet 6700, Thermo Scientific) with variable angle accessories (SpectraTech variable angle accessory and Harrick Scientific's 12° incident angle Specular Reflection Accessory). The spectral measurement results shown in Figures S2a and S2b are consistent with the direct-hemispherical reflectance shown in Figure 2c in the main text. The measured emissivity is nearly constant for all angles relevant for radiative transfer within the experimental setup (around normal direction due to the small spacing between the emitter, diffuse-solar-reflecting cover and aluminum aperture).

Figure S2. Measured wavelength- and angle-dependent emissivity. Measured spectral emissivity of the solar-white emitter (a) and solar-black emitter (b) at different angles of incidence. Average emissivity in the high-transparency atmospheric spectral window (8-13 μm) plotted as a function of incidence angle (c: solar-white emitter, d: solar-black emitter).

REVIEWERS' COMMENTS:

Reviewer #1 (Remarks to the Author):

The revision of this manuscript has filled in the requested missing details and required corrections and clarifications. It is now a far more readable and useful document involving a new idea in an energy and comfort related area of high current interest. The larger scale practicalities, engineering, installation, and prospects may limit applications to niche examples one being discussed later in the paper. Its predictions are well backed up by a thorough experimental study. As a new applied science initiative in an area of growing importance it should attract a lot of interest and I am now happy to see it published.

Reviewer #2 (Remarks to the Author):

As discussed in the prior review, this manuscript concerns using a polished aluminum disk to block solar radiation and create cooling of 5.5-6 K below ambient.

The authors have successfully addressed most of the points from the reviewers, with some modest exceptions. Below, I briefly summarize key points from each Reviewer, and comment on the degree to which these are addressed by the authors.

Reviewer 1

Practical usefulness: this shows how it is possible, but perhaps the reviewer was also interested in seeing a general analysis of the economic tradeoffs associated with a given application

Errors and error bars: the discrepancy between model and experiment is clear, but further comments on the generality of these differences would be appropriate here.

Additional references required: these are fully included now.

Reviewer 2

1. Reason for 50% difference in cooling power: the presence of high convection explains this result in a satisfactory manner; furthermore, the average convective coefficient is consistent with wind exposure.

2. Variation of convective coefficient: it is helpful that weather data was used to refine this value, and confirm that wind speed did not change significantly during the course of the experiment.

3. IR image of temperature distribution: comparison of thermocouple data with COMSOL model in Figure S5 is a satisfactory alternative to the original request

4. Combining CPV with directional radiative cooling: the illustrations in Figures S8 and S9 explain the concept clearly

Reviewer 3

1. Comparison to spectrally selective emitters: the authors make an interesting comparison here and this should be a valuable contribution to the literature.

2. Use in CPV systems: this mostly overlaps with Comment 4 from Reviewer 2

3. Energy balance missing a term for radiation from the emitter to reflector: this has now been corrected, and the reason it didn't cause a large error previously is clear.

4. Angular-dependence for emissivities of solar white and solar black emitters: the limited angular dependence makes sense, assuming that one is averaging both polarizations – this should be pointed out more clearly.

In conclusion, the authors have studied the effects of blocking the solar disc on radiative cooling, and shown some promising results for the resulting cooling power. They have successfully addressed most of the points raised by the reviewers. With some further modest changes discussed above, it should be appropriate to publish this manuscript in Nature Communications.

Reviewer #3 (Remarks to the Author):

The authors have addressed my concerns in the revised manuscript, and the study has become more readable. I now recommend its publication in Nature Communications.

Reviewer 1

REVIEWER

The revision of this manuscript has filled in the requested missing details and required corrections and clarifications. It is now a far more readable and useful document involving a new idea in an energy and comfort related area of high current interest. The larger scale practicalities, engineering, installation, and prospects may limit applications to niche examples one being discussed later in the paper. Its predictions are well backed up by a thorough experimental study. As a new applied science initiative in an area of growing importance it should attract a lot of interest and I am now happy to see it published.

RESPONSE

We thank the reviewer for providing useful suggestions which helped us improve the manuscript. We believe this work is an important contribution to the emerging field of passive radiative cooling.

Reviewer 2

REVIEWER

As discussed in the prior review, this manuscript concerns using a polished aluminum disk to block solar radiation and create cooling of 5.5-6 K below ambient. The authors have successfully addressed most of the points from the reviewers, with some modest exceptions. Below, I briefly summarize key points from each Reviewer, and comment on the degree to which these are addressed by the authors.

RESPONSE

We thank the reviewer for evaluating our revised manuscript and providing useful comments. We address each issue raised by the reviewer in our responses below. Based on the reviewers' suggestions, we performed additional measurements, revised modeling and included additional information to improve the manuscript. We hope these changes satisfactorily address the concerns of the reviewer.

REVIEWER

Reviewer 1

*Practical usefulness: this shows how it is possible, but perhaps the reviewer was also interested in seeing a general analysis of the economic tradeoffs associated with a given application.
Errors and error bars: the discrepancy between model and experiment is clear, but further comments on the generality of these differences would be appropriate here. Additional references required: these are fully included now.*

RESPONSE

We appreciate the reviewers' suggestions. The revised manuscript now highlights the general agreement between the model and experiment and clarifies that the ideal model described earlier in the paper was modified to model the performance of our particular system. The corresponding changes are included in the Experimental results section of the main text: "Figure 4b also plots the corresponding modeled device cooling performance which shows good agreement with experiment. The ideal model described earlier was modified to account for the measured spectral properties of the emitters, cover and reflector, device geometry, ambient temperature during the measurement, as well as the conductive-convective losses in the system."

We also carefully considered the reviewer's suggestion of possibly including an economic analysis of applications using the approach presented in this paper. Given that this is a first demonstration of the directional daytime radiative cooling concept, however, we concluded any commercial assessment would be premature and beyond the scope of this work. Thus we do not include further economic evaluation beyond the possible applications already outlined in Supplementary Note 9.

REVIEWER

Reviewer 2

- 1. Reason for 50% difference in cooling power: the presence of high convection explains this result in a satisfactory manner; furthermore, the average convective coefficient is consistent with wind exposure.*
- 2. Variation of convective coefficient: it is helpful that weather data was used to refine this value, and confirm that wind speed did not change significantly during the course of the experiment.*
- 3. IR image of temperature distribution: comparison of thermocouple data with COMSOL model in Figure S5 is a satisfactory alternative to the original request*
- 4. Combining CPV with directional radiative cooling: the illustrations in Figures S8 and S9 explain the concept clearly*

RESPONSE

We thank the reviewer for carefully assessing the changes made in the revised manuscript and for noting that no further changes are required for the mentioned comments.

REVIEWER

Reviewer 3

- 1. Comparison to spectrally selective emitters: the authors make an interesting comparison here and this should be a valuable contribution to the literature.*
- 2. Use in CPV systems: this mostly overlaps with Comment 4 from Reviewer 2*
- 3. Energy balance missing a term for radiation from the emitter to reflector: this has now been corrected, and the reason it didn't cause a large error previously is clear.*
- 4. Angular-dependence for emissivities of solar white and solar black emitters: the limited angular dependence makes sense, assuming that one is averaging both polarizations – this should be pointed out more clearly.*

RESPONSE

As per the reviewers' suggestion, the revised Supplementary Note 3 now clearly points out that the angular-dependent emissivity measured using FTIR spectrometer was "averaged over s- and p-polarizations". We thank the reviewer for this suggestion and for checking that all other comments were satisfactorily addressed.

REVIEWER

In conclusion, the authors have studied the effects of blocking the solar disc on radiative cooling, and shown some promising results for the resulting cooling power. They have successfully addressed most of the points raised by the reviewers. With some further modest changes discussed above, it should be appropriate to publish this manuscript in Nature Communications.

RESPONSE

We thank the referee for reviewing our work and providing helpful comments. We hope the changes highlighted in the responses above satisfactorily address the points raised by the reviewer, and that the revised manuscript is now ready for publication in *Nature Communications*.

Reviewer 3

REVIEWER

The authors have addressed my concerns in the revised manuscript, and the study has become more readable. I now recommend its publication in Nature Communications.

RESPONSE

We thank the reviewer for the positive assessment of the revised manuscript and for helping us improve the quality of our work.